# Correlates of Acquiring a Traumatic Brain Injury before Experiencing Homelessness: An Exploratory Study

**Stephanie Chassman** [1,*], **Katie Calhoun** [1] , **Blair Bacon** [2], **Sara Chaparro Rucobo** [3], **Emily Goodwin** [4], **Kim Gorgens** [3] and **Daniel Brisson** [3]

1 Graduate School of Social Work, University of Denver, Denver, CO 80208, USA
2 Anschutz School of Medicine, University of Colorado, Aurora, CO 80045, USA
3 Graduate School of Professional Psychology, University of Denver, Denver, CO 80210, USA
4 Rosemead School of Psychology, Biola University, La Mirada, CA 90639, USA
* Correspondence: stephanie.chassman@du.edu

**Abstract:** The rates of traumatic brain injury (TBI) are significantly higher among individuals experiencing homelessness compared to the general population. The relationship between TBI and homelessness is likely bi-directional as factors associated with homelessness may increase the risk of acquiring a TBI, and factors associated with TBI could lead to homelessness. This study builds upon previous research by investigating the following research questions: (1) What are the rates of TBI among a sample of individuals experiencing homelessness? (2) Does a TBI experience precede or follow an initial period of homelessness? And, (3) What are the correlates of TBI prior to homelessness including self-reported mental health variables? A cross-sectional study design and purposive sampling were utilized to interview 115 English-speaking adults (ages 18–73) in two Colorado cities. Results show, 71% of total participants reported a significant history of TBI, and of those, 74% reported a TBI prior to experiencing homelessness. Our logistic regression models reveal a significant relationship between mental health and acquiring a TBI prior to experiencing homelessness. Implications include prioritizing permanent supportive housing followed by other supportive services.

**Keywords:** homelessness; traumatic brain injury; mental health





## 1. Introduction

On a single night in January 2020, more than 580,000 people experienced homelessness in the United States (National Alliance to End Homelessness 2021). Compared to the previous year, homelessness increased by 2%, marking the fourth straight year of increases in homelessness nationwide (National Alliance to End Homelessness 2021). People experiencing homelessness are more likely to have various health issues such as diabetes, heart disease, and substance use (National Alliance to End Homelessness 2021), and are more likely to have reported a traumatic brain injury (TBI) at some point in their lifetime (Hwang et al. 2008; Oddy et al. 2012; Stubbs et al. 2020; Topolovec-Vranic et al. 2012).

A TBI is "an alteration in brain function, or other evidence of brain pathology, caused by an external force" (Menon et al. 2010, p. 1). There are several ways in which TBIs are classified, including the cause of impact, functional impairment, or physical change (Topolovec-Vranic et al. 2012). Several studies suggest that if one head injury has occurred, the likelihood of subsequent head injuries increases (Hwang et al. 2008; McMillan et al. 2015; Oddy et al. 2012).

The rates of TBI are significantly higher among individuals experiencing homelessness compared to the general population. Research has shown that more than half of individuals experiencing homelessness have sustained a TBI at some point in their lifetime; this is compared to the 12% lifetime prevalence rate among the general population (Hwang et al. 2008; Oddy et al. 2012; Stubbs et al. 2020; Topolovec-Vranic et al. 2012). Moreover, research

shows that over 60% of individuals experiencing homelessness with a history of TBI were found to have experienced more than one TBI (Hwang et al. 2008). Other research has shown that, in a quarter of the population of individuals experiencing homelessness with a TBI history, the injury was identified as moderate to severe (Boseley 2019).

## 2. Literature Review

### 2.1. Temporal Relationship between TBI and Homelessness

Individuals experiencing homelessness are at a disproportionately high risk for sustaining a TBI. Some circumstances increase the risk of sustaining a TBI including victimization by assaults, engagement in risky behavior, and higher rates of substance use (Backer and Howard 2007; Silver and Felix 1999). One study found that men who were chronically homeless with an alcohol-use problem had higher rates of head injuries (Svoboda and Ramsay 2014). Additional research has also shown that marijuana use, and crack or cocaine use was common among TBI participants (Mackelprang et al. 2014). Substance use increases both the risk of homelessness and the risk of TBI (Corrigan 1995; Hwang et al. 2008). As a result, individuals experiencing homelessness have an increased susceptibility to brain injury, suggesting that homelessness may be a contributing factor to the increased rates of head injuries.

Alternatively, among several studies including homeless populations, the majority of participants reported their first TBI prior to becoming homeless, suggesting that TBI may be a risk factor for homelessness (Hwang et al. 2008; Mackelprang et al. 2014; Oddy et al. 2012; Topolovec-Vranic et al. 2014). TBI is associated with low subsequent employment rates (approximately 40%), and memory issues and difficulties in planning may affect the ability of someone to maintain employment and housing (Oddy et al. 2012) which often contributes to a downward spiral into homelessness (Rogers and Read 2007; Topolovec-Vranic et al. 2012; Van Velzen et al. 2009). TBI add barriers to maintaining stable housing; for instance, behavioral issues resulting from TBI may be mistaken for non-compliance by landlords and neighbors and thus impact an individual's ability to budget, pay rent, and maintain a home (Roach 2016).

Barnes et al. (2015); Figure 1 posited the bi-directional relationship between TBI and homelessness. Those authors argue this relationship is likely bi-directional since factors associated with homelessness may increase the risk of acquiring a TBI, and factors associated with TBI could make an individual vulnerable to becoming homeless. It is important to examine this bi-directional relationship as it pertains to pathways into and out of homelessness and the impact of TBI on this population. Determining the timeline of brain injury history in relation to the onset of homelessness is an aspect of this relationship that merits further investigation.

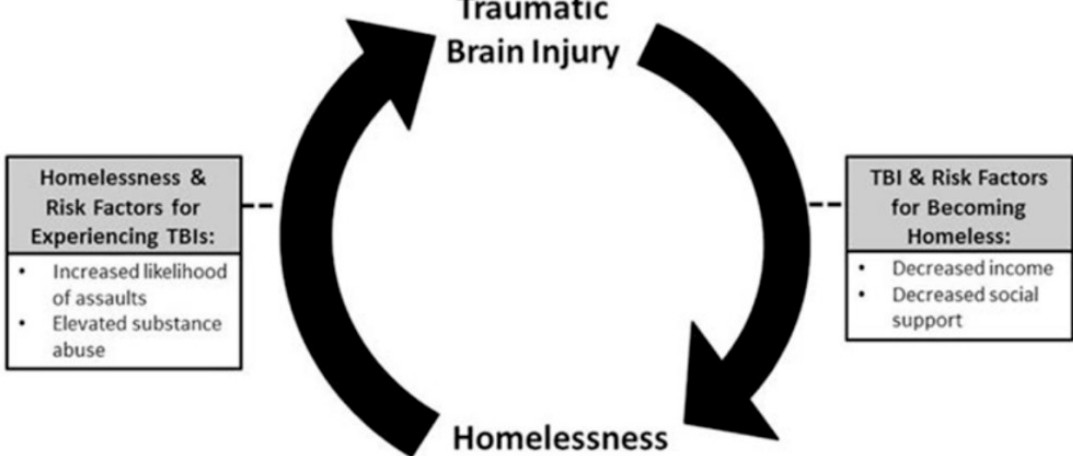

**Figure 1.** Bi-Directional Relationship Between TBI and Homelessness (Barnes et al. 2015).

### 2.2. Co-Morbidities of TBI and Homelessness

2.2.1. Mental Health and Substance Use

Mental health and substance use are associated with both TBI and homelessness. The few studies that have reviewed specific associations of TBI, homelessness, and a mental health diagnosis found linkages to major depressive episodes, PTSD, panic disorder, mood disorder with psychotic features, psychotic disorders (Topolovec-Vranic et al. 2017), depression, anxiety, and bipolar disorder (Palladino et al. 2017). While TBI may increase the risk of subsequent mental health problems, it is also likely that pre-existing mental health and substance use problems increase the risk of TBI (Hwang et al. 2008). It is difficult for studies to claim intersections of TBI, homelessness, and a specific mental health diagnosis as it is challenging to assess if mental health symptoms are true mental health problems or consequences of TBI that may appear as mental health issues. The direction and correlates, such as mental illness, of the relationship between TBI and homelessness should be further studied.

A history of TBI among individuals experiencing homelessness has been associated with substance use. Research has found support for a relationship between substance use and other risky behaviors and injuries, including TBI (Brenner et al. 2017). Substance use increases both the risk of homelessness and the risk of TBI (Corrigan 1995; Hwang et al. 2008). Substance use can decrease opportunities to establish and maintain housing and employment, and increase exposure to victimization (Dunne et al. 2015), which may lead one to sustaining a brain injury. Additionally, substance use may have deleterious effects on recovery from a TBI (Corrigan 1995), potentially also exacerbating the effects of subsequent head injuries.

2.2.2. Barriers to Housing

TBI can impede individuals' housing stability. Adshead et al. (2019) explained that consequences from a TBI such as executive impairments, paired with challenges to engaging in social care, increase the risk for homelessness through an inability to receive the support necessary to remain in one's home and pay the bills. Individuals' cognitive impairments may make it difficult to understand how to pay rent, utilities, and maintain their residences (HCH Clinician's Network 2003). TBI can impact one's housing stability making it difficult to maintain housing.

Pathways into homelessness often include economic factors that interact with mental health factors, which include substance use, mental health problems, and family conflict (Dykeman 2011; Harrington 1985; Sullivan et al. 2000). Individuals with mental illness are over-represented among homeless populations compared to the general population (Hux et al. 2009; Hwang et al. 2008; Mackelprang et al. 2014; Svoboda and Ramsay 2014; To et al. 2015; Topolovec-Vranic et al. 2014). Mental illness has been found to be a risk factor that contributes to homelessness (Sullivan et al. 2000). These individuals may become homeless due to the sequalae of their mental health and/or as a consequence of broken social ties and economic networks (Sullivan et al. 2000). Additionally, homelessness can precipitate and worsen mental illness, both independent of and in the context of substance use (Bresnahan et al. 2003). Furthermore, individuals experiencing homelessness suffering from a mental illness face a myriad of health and social problems. Individuals with severe mental illness who become homeless are also at a higher risk of long-term homelessness (Smartt et al. 2019).

Research has shown that individuals experiencing homelessness with a history of TBI are more likely to have a history of mental health disorders (Hux et al. 2009; Hwang et al. 2008; Mackelprang et al. 2014; Svoboda and Ramsay 2014; To et al. 2015; Topolovec-Vranic et al. 2014) and worse scores on quality of life screening measures (Hwang et al. 2008; To et al. 2015).

### 2.3. The Current Study

While previous studies provide important and critical information on the comorbidity between TBI and homelessness, more research is needed to examine the temporal

relationship as it pertains to pathways into and out of homelessness. Furthermore, the direction and correlates, such as mental illness, should be further studied as this represents a gap in the literature. This paper seeks to fill this gap by examining the correlates of TBI and homelessness.

This study expands on previous research by describing with greater specificity, the rates and directionality, along with mental health correlates of TBI and homelessness.

Research questions include:

1. What are the rates of TBI among a sample of individuals experiencing homelessness?
2. Does a TBI experience precede or follow an initial period of homelessness?
3. What are the correlates of TBI prior to homelessness including self-reported mental health variables?

## 3. Materials and Methods

### 3.1. Study Setting

In 2020, researchers from The University of Denver (the Center for Housing and Homelessness Research and the Graduate School of Professional Psychology) partnered with two community organizations across Colorado serving individuals experiencing homelessness. The dataset came from a two-site study (The Murphy Center for Hope in Fort Collins and Catholic Charities' Marian House in Colorado Springs) examining the relationship between TBI and homelessness.

### 3.2. Study Design

A cross-sectional study design and purposive sampling across two sites were utilized to interview a total of 115 English-speaking adults (ages 18–73). Quantitative questions examined rates of TBI, experiences of homelessness, time sequencing of TBI and homelessness events, and correlates of housing instability including self-reported mental health variables.

### 3.3. Sample and Recruitment

Community partners supported the study by hanging recruitment flyers in service provision areas in their offices as well as encouraging service recipients to visit on the day of the data collection.

A standardized protocol for recruiting and screening potential participants was used across the sites. The eligibility screener assessed if a participant was over 18 years old and experiencing homelessness or was in unstable housing, from a self-report. Written informed consent was given by eligible participants before beginning data collection. The Institutional Review Board (IRB) at the University of Denver approved all study procedures prior to data collection.

### 3.4. Data Collection Procedures

Participants were given the written consent form and asked if they would like to read it themselves or have it read to them. Once written consent was obtained, researchers read each survey question to participants and allowed participants to answer. Participants were told they could skip any questions they were uncomfortable answering and trained staff were available for support. The survey took approximately 25 min to complete. Participants were given a $15 gift card for a local grocery store as compensation for survey completion.

### 3.5. Measures

3.5.1. Sociodemographic Characteristics

Sociodemographic characteristics included the following variables: Study site (Fort Collins or Colorado Springs). Gender identity was originally measured using three categories (male, female, other—specify). Since a majority of participants identified as either male or female (98.8% of the sample), a dichotomous variable identified participants as male or female, those who did not identify as gender binary were dropped from the analysis. Sexual orientation was originally categorized into five categories (bisexual, gay,

heterosexual, lesbian, not listed). Similarly, 85.2% of the sample identified as heterosexual, therefore a dichotomous variable identified participants as heterosexual and LGB or not listed. Race was originally categorized into eight categories (American Indian/Alaska Native, Asian, Native Hawaiian or Other Pacific Islander, Black or African American, White, Hispanic, more than one race, unknown/not reported) and recoded into two categories (white and BIPOC) for analysis, due to the limited representation in some subgroups. Levels of education were originally measured using the following: less than a high school diploma; high school degree or equivalent; Associate degree; Bachelor's degree; Master's degree; Doctorate; other. These were recoded into two categories (high school degree or less and more than high school degree), similarly, due to the limited representation in some subgroups (e.g., Master's degree, Doctorate).

Two standardized measures were used to assess homelessness status and TBI history. The Vulnerability Index, Service Prioritization Decision Assistance Tool (VI-SPDAT; Community Solutions 2015) was used to assess homelessness status and the Ohio State University TBI Identification Method (OSU TBI-ID; Corrigan and Bogner 2007) was used to assess TBI history.

### 3.5.2. VI-SPDAT

The VI-SPDAT is the homelessness status tool used by the Continuum of Care (COC) directed by the Department of Housing and Urban Development (HUD) to assess homelessness status, to prioritize which clients should receive housing assistance first (Community Solutions 2015). The VI-SPDAT was used to assess history of housing and homelessness, risk behavior, socialization, and daily functioning and wellness.

### 3.5.3. Mental Health and Housing Stability (Dependent Variables)

The impact of mental health on housing stability was measured through the VI- SPDAT, specific questions included: Have you ever had trouble maintaining your housing, or been kicked out of an apartment, shelter program or other place you were staying, because of a mental health issue or concern? And, do you have any mental health or brain issues that would make it hard for you to live independently because you would need help? (1 = yes, 0 = no).

### 3.5.4. OSU TBI-ID

The OSU TBI-ID (Corrigan and Bogner 2007) was used to collect information on participants' history and experiences with TBI. The OSU TBI-ID is a standardized structured interview procedure designed to elicit reports of lifetime TBI history from participants. Participants were considered to have a significant history of TBI if they reported a "first" TBI with loss of consciousness (LOC) before the age of 15, a "worst" TBI with LOC longer than 30 min, or a "multiple"TBI event, defined as "a period where three or more blows to the head caused altered consciousness OR two or more TBIs with LOC within a 3-month period" (Glover et al. 2018, p. 16). For analysis, scores of "first" "worst," or "multiple" were utilized and if a participant screened positive for any of the criteria, they were scored as having a reported TBI (1 = yes, 0 = no). Additionally, the OSU TBI-ID measures mechanism of injury as the following: car accident or from crashing some other moving vehicle such as a bicycle, motorcycle or ATV; being hit by something (for example, falling from a bike or horse, rollerblading, falling on ice, being hit by a rock or playing sports or on the playground); being hit by someone, shaken violently or been shot in the head; nearby when an explosion or blast occurred.

To determine whether a TBI came before or after homelessness, the age at which a participant first reported a TBI was compared to the age a participant reportedly first experienced homelessness. If the age of TBI was younger than the age when they were first homeless, it was coded "TBI first." If the age of homelessness was younger than the age of the first reported TBI, it was coded "homelessness first". Participants who did not

have a reported TBI or who had a reported TBI at the same age they first experienced homelessness were excluded.

### 3.6. Analytic Approach

Study data were collected and managed using REDCap electronic data capture tools hosted at The University of Denver. REDCap (Research Electronic Data Capture; Harris et al. 2009) is a secure, web-based application designed to support data capture for research studies, providing: (1) an intuitive interface for validated data entry; (2) audit trails for tracking data manipulation and export procedures; (3) automated export procedures for seamless data downloads to common statistical packages; and (4) procedures for importing data from external sources. Listwise deletion was utilized for missing data because less than 10% of the data were missing.

Data analysis was conducted using the Statistical Package for the Social Sciences (SPSS; version 26; IBM Corp, 2017). First, a series of descriptive analyses were conducted to describe sample characteristics in terms of TBI variables as well as all independent variables. Dependent variables were chosen based on our review of the previous literature, specifically the literature that examines the relationship between mental health and housing stability (Hux et al. 2009; Hwang et al. 2008; Mackelprang et al. 2014; Sullivan et al. 2000; Svoboda and Ramsay 2014; To et al. 2015; Topolovec-Vranic et al. 2014). The sample characteristics were described for the full sample.

Binary logistic regression analysis was then conducted by regressing the two-category mental health dependent variables on the independent variables, which include demographic and TBI first variable; parameter estimates from the logistic regression model are presented as odds ratios (ORs) alongside 95% confidence intervals, and *p*-values. Additionally, the analysis includes adjusted ORs for all significantly associated independent variables. There were two dependent variables: Have you ever had trouble maintaining your housing, or been kicked out of an apartment, shelter program or other place you were staying, because of a mental health issue or concern? And, do you have any mental health or brain issues that would make it hard for you to live independently because you would need help? (1 = yes, 0 = no).

## 4. Results

Fifty-nine participants completed the survey on 3 March and 56 participants on 2 October 2020, for a total of 115 participants. Out of the 115 participants, 85 (74%) participants had reported a TBI prior to experiencing homelessness for the first time. Quantitative descriptive characteristics, homelessness-related variables, and TBI-related variables for the 85 participants who had reported a TBI prior to experiencing homelessness are reported in Table 1.

**Table 1.** Descriptive Characteristics of Participants with a Reported TBI Before Experiencing Homelessness.

| Descriptive Characteristics of Participants with a Reported TBI before experiencing homelessness (N = 85) | |
| --- | --- |
| **n (%) or M (SD)** | |
| **Gender** | |
| Male | 56 (65.9) |
| Female | 28 (32.9) |
| **Sexual Orientation** | |
| Heterosexual | 72 (84.7) |
| Not heterosexual | 13 (15.3) |

**Table 1.** *Cont.*

| | |
|---|---|
| **Race and Ethnicity** | |
| White | 52 (61.2) |
| Not white | 33 (38.8) |
| **Education** | |
| High school degree or less | 54 (63.5) |
| More than high school degree | 31 (36.5) |
| **Homelessness Variables** | |
| Sleep in shelters | 45 (52.9) |
| Sleep outside | 27 (31.8) |
| Other | 9 (10.6) |
| Transitional housing | 4 (4.7) |
| No. episodes of homelessness (M, SD) | 3.5 (4.4) |
| **Age (years) (M, SD)** | 31 (13) |
| **Housing Stability-Mental Health** | |
| Have you had trouble maintaining housing due to a mental health issue or concern (1 = yes) | 16 (18.8) |
| Do you have a mental health or brain issue that would make it hard for you to live independently? (1 = yes) | 14 (16.5) |
| Do you have planned activities that make you feel happy and fulfilled? (0 = no) | 23 (27.1) |
| **TBI Variables for Total Sample (n = 115)** | |
| "Worst" injury (LOC for more than 30 min) | 53 (46.1) |
| "First" injury (LOC before age 15) | 28 (24.3) |
| "Multiple" injury (three or more head injuries resulting in an altered state or two or more TBIs with LOC within a 3-monthperiod | 57 (49.6) |
| **Mechanism of TBI** | |
| Fall-related injury | 46 (55) |
| Motor vehicle accident | 45 (53) |
| Assault | 45 (53) |
| Near a blast or explosion | 18 (21) |

### 4.1. Descriptive Characteristics of Participants with a Reported TBI before Experiencing Homelessness

#### 4.1.1. Descriptive Characteristics

Out of the 85 participants from Fort Collins and Colorado Springs who sustained a TBI prior to experiencing homelessness, 66% (*n* = 56) identified as male, and 33% (*n* = 28) as female. Regarding sexual orientation and race, most participants identified as heterosexual (85%, *n* = 72), and 61% (*n* = 52) identified as white. In terms of education, 63% (*n* = 54) of participants had completed a high school degree or less.

#### 4.1.2. Homelessness Variables

Regarding homelessness characteristics from the VI-SPDAT, 53% (*n* = 45) reported that they currently slept in shelters most frequently, followed by 32% (*n* = 27) who slept outdoors, 10% (*n* = 9) who slept in other-non listed locations, and 4% (*n* = 4) who slept in transitional housing. On average, participants who sustained a TBI before experiencing homelessness were 31 years old (SD = 13) and had experienced homelessness an average of 3.5 times (SD = 4.4).

#### 4.1.3. Brain Injury

Out of the 115 total participants, 71% reported a significant history of TBI; the remaining participants did not report a history of TBI. Out of the participants who reported a history of TBI, 74% (*n* = 85) had a reported TBI prior to their first experience of homelessness. The OSU TBI-ID screening revealed that 46% (*n* = 53) of participants reported at

least one head injury with a LOC for more than 30 min (worst). More than 24% (*n* = 28) of participants reportedly experienced a TBI with LOC before the age of 15 (first). Additionally, almost 50% (*n* = 57) of participants reported experiencing either three or more head injuries resulting in an altered state, or two or more TBIs with LOC within a 3-month period (multiple). Regarding the reported mechanisms of participants' TBI, 55% (*n* = 46) of participants experienced a fall-related injury, 53% (*n* = 45) reported a motor vehicle accident, 53% (*n* = 45) of participants reported an assault, and 21%(*n* = 18) were near an explosion or blast.

### 4.1.4. Mental Health (Dependent Variables)

When examining mental-health-related variables, participants who sustained a TBI prior to experiencing homelessness reported that mental health was a contributing factor toward housing instability. Specifically, 19% (*n* = 16) of participants reported they had trouble maintaining their housing, or had been kicked out of an apartment, shelter program or other place they were staying because of a mental health issue or concern. And, 16% (*n* = 14) reported that they had a mental health or brain issue that would make it hard to live independently because they would need help. Additionally, of those participants who said yes, they do have mental health or brain issues that would make it hard to live independently because they would need help, 14 (19%) identified as white, and 10 (24%) identified as not white, 22 (23%) identified as heterosexual, and 2 (12%) identified as not heterosexual.

Of those who answered yes, they have had trouble maintaining housing, or been kicked out of an apartment, shelter program or other place they were staying, because of a mental health issue or concern, 19 (26%) identified as white, 10 (24%) identified as not white, 24 (25%) identified as heterosexual, and 5 (29%) identified as not heterosexual.

### 4.2. Impact of TBI Prior to Homelessness and Mental Health

Multivariable findings are presented in Tables 2 and 3. There were two outcomes of interest relating to mental health: one measure of mental health challenges impacting housing and one measure of mental health. While demographic variables were not significantly associated with either mental health dependent variables, the TBI first variable was. Specifically, for participants who had a reported TBI prior to experiencing homelessness, they were less likely to have experienced housing instability due to a mental health issue or concern (OR = 0.27, *p* < 0.01, CI = 0.11, 0.67). Additionally, for participants who had a reported TBI prior to experiencing homelessness, they were less likely to have a mental health or brain issue that would make it hard for them to live independently (OR = 0.36, *p* < 0.05, CI = 0.13, 0.96).

**Table 2.** Multivariable Findings.

| Do you have any mental health or brain issues that would make it hard for you to live independently because you would need help? | | |
|---|---|---|
| **Factors** | OR | 95% CI |
| Race/Ethnicity | 0.63 | 0.24–1.65 |
| Gender | 0.66 | 0.24–1.8 |
| Sexual Orientation | 2.53 | 0.48–13.3 |
| Education | 1.75 | 0.63–4.89 |
| TBI before Homelessness | 0.36 * | 0.13–0.96 |

Note. Reference category for: Do you have any mental health or brain issues that would make it hard for you to live independently because you would need help? (0 = no). OR = odds ratio; 95% CI = 95% confidence interval. * *p* < 0.05. ** *p* < 0.01 *** *p* < 0.001.

**Table 3.** Multivariable Findings.

| Have you ever had trouble maintaining your housing, or been kicked out of an apartment, shelter program or other place you were staying, because of: A mental health issue or concern? | | |
| --- | --- | --- |
| **Factors** | OR | 95% CI |
| Race/Ethnicity | 0.94 | 0.37–2.39 |
| Gender | 0.92 | 0.35–2.3 |
| Sexual Orientation | 0.74 | 0.21–2.56 |
| Education | 1.67 | 0.62–4.45 |
| TBI before Homelessness | 0.27 ** | 0.11–0.69 |

Note. Reference category for: Have you ever had trouble maintaining your housing, or been kicked out of an apartment, shelter program or other place you were staying, because of a mental health issue or concern? (0 = no). OR = odds ratio; 95% CI = 95% confidence interval. * $p < 0.05$. ** $p < 0.01$. *** $p < 0.001$.

## 5. Discussion

This study aimed to address three research questions: What are the rates of TBI among a sample of individuals experiencing homelessness? Does a TBI experience precede or follow an initial period of homelessness? And, what are the correlates of experiencing a TBI prior to homelessness including self-reported mental health variables? Several significant findings emerged from this study that broaden our knowledge of the correlates of TBI preceding homelessness among a sample of adults experiencing homelessness.

Overall, 71% of total participants reported a significant history of TBI, and of those, 74% reported that their TBI occurred before experiencing homelessness. In comparison, research indicates that approximately half of individuals experiencing homelessness have a TBI history (Boseley 2019; Hwang et al. 2008; Russell et al. 2013). This study found higher rates of reported TBI among a sample of adults experiencing homelessness compared to what the extant literature suggests, further suggesting that TBI may be a significant risk factor of homelessness. Moreover, these data were collected during the COVID-19 pandemic, perhaps showing the impact of COVID-19 on homelessness and brain injuries. More research is needed to examine the relationship between homelessness and TBI and the impact that COVID-19 had on housing stability.

Our logistic regression models revealed a negative relationship between mental health variables and a reported TBI prior to experiencing homelessness. Specifically, for participants who had a reported TBI prior to experiencing homelessness, they were less likely to have experienced housing instability due to a mental health issue or concern, and they were less likely to have a mental health or brain issue that would make it hard for them to live independently. These important and counterintuitive findings may suggest that the relationship between TBI and homelessness is perhaps driven by other factors related to the injury including health issues, cognitive impairment, substance use, and victimization. Additionally, structural level factors such as a lack of affordable housing, loss of a job, eviction, domestic violence, medical debt, lack of insurance, and income inequality are associated with homelessness (Burt and Aron 2001; Dykeman 2011; Shinn 2007) and may also be associated with TBI. More research is needed to understand the impact of structural-level factors on housing instability and TBI.

A history of TBI has been found to be strongly associated with several adverse health outcomes among individuals experiencing homelessness, including seizures (Hwang et al. 2008), cognitive impairment (Gicas et al. 2020), back problems, chronic hepatitis, migraine headaches, arthritis, lung disease, high blood pressure, HIV or AIDS (Nikoo et al. 2014), concentration difficulties, excessive worry or difficulty sleeping (Mackelprang et al. 2014), and epilepsy (Topolovec-Vranic et al. 2017). More research is needed to evaluate the health problems and cognitive impairments associated with TBI which may contribute to the chronicity of homelessness and risk of subsequent TBI.

Additionally, research has found support for a relationship between substance use and other risky behaviors and injuries, including TBI (Brenner et al. 2017). Substance use increases the risk of TBI (Corrigan 1995; Hwang et al. 2008). And, substance use may have deleterious effects on recovery from a TBI (Corrigan 1995), potentially exacerbating

subsequent head injuries. Additionally, data have shown that victims of a TBI often reported alcohol intoxication at the time of injury (Corrigan 1995), further demonstrating that substance use increases the risk of TBI. More research is needed to evaluate substance use as a risk factor for TBI among individuals experiencing homelessness.

Furthermore, research shows that individuals experiencing homelessness experience high levels of stress and victimization (Robinson 2010; Toro et al. 2008). The stress of victimization and stigmatization may put one at higher risk of acquiring a TBI. Robinson (2010) further states that the longer a period of homelessness is prolonged, particularly when complicated by mental health, the more likely it is for a person to experience repeated victimization, potentially leading to brain injuries.

*5.1. Implications*

This study found that 71% of total participants reported a significant history of TBI, and of those, 74% reported that their TBI occurred before experiencing homelessness. Prevention strategies are recommended to prevent individuals with head injuries from experiencing housing instability and homelessness. It is recommended that individuals with a TBI be referred to social support groups, money management classes, and job training to establish and/or maintain stable housing while recovering from a TBI. These prevention strategies should take place in medical and clinical settings and include other individuals with TBI to increase social support. Prevention efforts should also emphasize low barriers to housing services to reduce isolation associated with homelessness. Additionally, partnerships across disciplines and organizations are recommended to improve outcomes for individuals experiencing homelessness (Topolovec-Vranic et al. 2013).

As previously mentioned, this study shows a negative relationship between having a TBI prior to experiencing homelessness and mental health variables that may suggest the relationship between TBI is driven by other risk factors related to the injury. More research is needed to further study the risk factors related to TBI and homelessness. Increasing knowledge and awareness among caregivers and medical professionals may lead to more TBI screening for those seeking medical care following a head injury; more screening for TBI may aid in accurate diagnosis to determine if an individual has a TBI, mental health issues, or both.

The comorbidity of TBI and mental health conditions can also create a challenge in diagnosing and treating TBIs in the unhoused population. An initial diagnosis of a TBI typically relies on self-reporting of specific symptoms at the time of injury, which can be missed or challenging to gather (National Academies of Sciences, Engineering, Medicine, & National Research Council 2019). This study used the OSU TBI-ID, which gathers information about TBI history across a person's lifespan. Because TBIs are so prevalent among people experiencing homelessness, the OSU TBI-ID can be an important screening tool for practitioners to use in conjunction with mental health screening instruments to gather information about the history of TBIs and separate out symptoms from a TBI from symptoms from a mental health condition —as many symptoms are the same. Incorporating the OSU TBI-ID into intake assessments or other health assessments can help practitioners identify TBIs and recommend additional assessment.

TBI can impede individuals' housing stability (Adshead et al. 2019; HCH Clinician's Network 2003). The mental health variables that we used in this study specifically explored how mental health impacted housing stability. We found that people who reported their first TBI prior to experiencing homelessness were less likely to have issues with housing stability due to a mental health condition. With this in mind, programs that prioritize housing, such as those that use a housing first model, can allow individuals to be placed into permanent housing and then focus on other treatment that may have contributed to housing instability.

*5.2. Limitations*

Certain study limitations should be noted. The study results are based on cross-sectional data. Future research on examining the correlates of TBI before homelessness may consider longitudinal data to draw causal relationships. Furthermore, the mental-health-related dependent variables used in this study came from the VI-SPDAT, a measure designed to screen for housing assistance. Future studies should consider using validated mental health measures to assess the impact of TBI and homelessness on mental health and housing stability. Additionally, the data used were self-reported and could be biased due to the sensitive topics asked of participants. Further, participants were recruited from service agencies serving adults experiencing homelessness, so the sample is likely not representative of all adults experiencing homelessness, especially those who are disconnected from services. Furthermore, individuals who did not identify as gender binary were dropped from the analysis. Future research should consider using diverse samples of participants to better understand the impact of gender identity on homelessness and TBI-related outcomes. Also, this study was limited geographically to two cities in Colorado, and while the demographic information is largely reflective of the demographics of the city and state, additional research should consider using probability samples from diverse geographic areas to obtain more generalizable findings. Furthermore, this study only included English-speaking adults. Researchers should consider administering their study in multiple languages in the future.

## 6. Conclusions

These findings demonstrate that the rates of reported TBI and rates of reported TBI prior to experiencing homelessness are high. Our findings suggest that participants with a reported TBI prior to homelessness were less likely to have housing instability related to a mental health concern. While our findings are preliminary, they offer important implications for intervention efforts when assisting individuals experiencing homelessness. In particular, given the prevalence of reported TBIs among our sample of individuals who are unhoused, it may be prudent to expand screening for TBI in medical or clinical settings among individuals experiencing homelessness. In addition, because of the prevalence rates of mental health variables, intervention and prevention efforts aimed at addressing homelessness should focus on mental health screening and treatment. Additionally, homelessness is but one at-risk TBI population; individuals experiencing domestic violence, incarceration, and substance use may also benefit from increased TBI screenings, as well as intervention and prevention efforts.

**Author Contributions:** Conceptualization, S.C., B.B., E.G., K.G. and D.B.; Formal analysis, S.C.; Investigation, S.C.; Methodology, S.C.; Project administration, S.C., K.G. and D.B.; Supervision, K.G. and D.B.; Writing—original draft, S.C., K.C., B.B., S.C.R. and E.G.; Writing—review & editing, S.C., K.C., B.B., S.C.R., K.G. and D.B. All authors have read and agreed to the published version of the manuscript.

**Funding:** This research was funded by the University of Denver Professional Research Opportunity for Faculty. And by Mindsource Brain Injury Network in Colorado.

**Institutional Review Board Statement:** The study was conducted in accordance with the University of Denver, and approved by the Institutional Review Board of University of Denver (1521142-9 and 10/19/21).

**Informed Consent Statement:** Informed consent was obtained from all subjects involved in the study.

**Data Availability Statement:** Not applicable.

**Conflicts of Interest:** The authors declare no conflict of interest.

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
