# Peer review of "Correlates of Acquiring a Traumatic Brain Injury before Experiencing Homelessness: An Exploratory Study"

_socsci, doi:10.3390/socsci11080376_

Round 1

Reviewer 1 Report

Correlates of Acquiring a Traumatic Brain Injury Before Experiencing Homelessness: An Exploratory Study is an excellent study that should be published. I think, however, that the presentation of responses from the OSU TBI structured interview assessment tool needs to provide clearer explanation of the subjects' experiences of TBI. In Step 1 of question 1-5, the OSU TBI interview tool requests that the cause be given. Readers of this study are going to be interested in those sources of exposure. How many are car crashes? How many are sports injuries? And the like.

Beyond that, readers are going to be interested in learning whether there are patterns and associations between sources of prior exposure and homelessness. Are the authors able to distinguish between causes of exposure prior to homelessness and experiences of exposure after homelessness? In short, the authors need to worker harder to show the descriptive results of their 115 subjects on the OSU TBI interview sheet and then also look for statistical relationships to prior and post homelessness. That needs to be done for (a) sources of exposure, (b) repeated occurrences, and (c) severity. Reading this paper, I would hazard a guess that a history of sports-related concussions and a single severe injury would be a pattern of repeated exposure that preceded homelessness, but that individuals reporting no prior experience would have had more severe injuries after they became homeless. This paper does not really provide me with any capacity to study whether any of that is true about the mechanism of exposure, and because the literature reviews sets-up that bi-directional relationship, it makes complete sense that readers might desire to analyze that data. We should be given the opportunity. What we have is only fairly descriptive data generally but the OSU questionnaire allows more detailed presentation.

This paper would also be improved by noting that homelessness is but one at risk TBI population, and that it is a condition that overlaps with other experiences that include intimate partner violence, incarceration, and addiction. Bringing this study into conversation with Eve Valera's work on IPV or Homer Venter's studies of inmates in NYC jails would extend the value of this study and likely its readership base. There is also historical scholarship on marginal populations and traumatic brain injury that would help readers understand the longevity of this social problem (see ST Casper and K O'Donnell 2020 or ST Casper 2022).  

Reviewer 2 Report

Abstract: Please detail what TBI stands for

3.6 Analytic approach:  Please add that parameter estimates from the logistic regression model will be presented as odds ratios (ORs) alongside 95% confidence intervals, and p-value.

Also please explicitly state what the two dependent, self-report mental health variables were. Its not obvious from section 3.5.3 (although they are stated in tables 1.2/1.3).

Results

Table 1.1  - some clarifications required:

·       Please add a column for the n=30 that experienced a TBI after experiencing homelessness and also a total column (seeing as TBI is reported for all n=115).

·       Please indicate which variables use a mean (sd) as its not always clear.

·       Please indicate which variables use VI-SPADT. Under ‘Homelessness Variables’ – is this historical or current status?

·       With the mental health housing stability measure, does responding yes to any of these three questions indicate a problem?

·       you state in the text that 71% reported a significant history of TBI-please add the number that had no TBI and those that had at least one (assuming that individuals can be in more than one category?). DO you think its worth reporting combinations (e.g. worst/first)?

·       Please add the definitions of First, Worst, Multiple to the table.

·       Please add the data on mechanisms of TBI to the table.

Section 4.2 and table 1.2/1.3 - Please change ‘Multivariate’ to ‘Multivariable’.

Tables 1.2/1.3

·       Please add the numbers and %  of the factors by presence or absence of the response to the mental health dependent variables, including reference categories for all independent variables (e.g. race, gender). This helps the reader understand what the numbers are involved and also the comparator categories.

·       Assuming that these are adjusted ORs (so each OR estimate has been adjusted for the other variables in the model), I would also add crude ORs (so looking at individual relationship between DV and IV). Sometimes ORs can be distorted in a multivariable analysis.  Please also add this detail to the methods section.

·       Please do not use the notation *, **, *** to indicate p values. Please add a column with the actual p-value.

General comments: Please add numbers to accompany % in the text.
